# Investigation of the iCC Framework Performance for Solving Constrained LSGO Problems [†]

## Alexey Vakhnin [1] and Evgenii Sopov [1,2,*]

[1] Department of System Analysis and Operations Research, Reshetnev Siberian State University of Science and Technology, Krasnoyarsky Rabochy Av. 31, 660037 Krasnoyarsk, Russia; alexeyvah@gmail.com
[2] Department of Information Systems, Siberian Federal University, Svobodny Av. 79, 660041 Krasnoyarsk, Russia
[*] Correspondence: evgenysopov@gmail.com
[†] This paper is an extended version of our paper published in IOP Conference Series: Materials Science and Engineering, II International Scientific Conference—MIST: Aerospace—2019: Advanced Technologies in Aerospace, Mechanical and Automation Engineering, 18–21 November 2019, Krasnoyarsk, Russia.

**Abstract:** Many modern real-valued optimization tasks use "black-box" (BB) models for evaluating objective functions and they are high-dimensional and constrained. Using common classifications, we can identify them as constrained large-scale global optimization (cLSGO) tasks. Today, the IEEE Congress of Evolutionary Computation provides a special session and several benchmarks for LSGO. At the same time, cLSGO problems are not well studied yet. The majority of modern optimization techniques demonstrate insufficient performance when confronted with cLSGO tasks. The effectiveness of evolution algorithms (EAs) in solving constrained low-dimensional optimization problems has been proven in many scientific papers and studies. Moreover, the cooperative coevolution (CC) framework has been successfully applied for EA used to solve LSGO problems. In this paper, a new approach for solving cLSGO has been proposed. This approach is based on CC and a method that increases the size of groups of variables at the decomposition stage (iCC) when solving cLSGO tasks. A new algorithm has been proposed, which combined the success-history based parameter adaptation for differential evolution (SHADE) optimizer, iCC, and the $\varepsilon$-constrained method (namely $\varepsilon$-iCC-SHADE). We investigated the performance of the $\varepsilon$-iCC-SHADE and compared it with the previously proposed $\varepsilon$-CC-SHADE algorithm on scalable problems from the IEEE CEC 2017 Competition on constrained real-parameter optimization.

**Keywords:** large-scale global optimization; evolution algorithms; differential evolution; cooperative coevolution; constrained optimization

---

## 1. Introduction

The progress of human activity in different areas does not stand still and optimization problems are no an exception as they have become more complex. For today, the general optimization task is a "black-box" optimization problem (there is no knowledge in an explicit form about functional dependence of variables) and it is high dimensional (uses more than hundreds/thousands of variables). Such problems are referred to as large-scale global optimization (LSGO) problems [1]. In the scientific publication [2], the current state of LSGO was discussed. In fact, tasks with a large number of objective variables are a challenging task for a wide range of optimization techniques and algorithms. A typical LSGO is single-objective and boxed constrained (variables are constrained only by lower and upper boundaries). In addition, many real-world optimization tasks exist which are constrained by functions (equalities and inequalities) [3,4]. The constrained optimization problems (COPs) have additional

requirements, as opposed to unconstrained optimization problems that separate the search space into feasible and infeasible domains. The majority of modern COPs are dealing with BB objective functions and BB constraints. In other words, we can only evaluate objective functions and constraints in candidate solutions. Constrained LSGO (cLSGO) problems have not been well studied yet. The cLSGO problem can be formulated in the following way:

$$f(\overline{x}) \to \min_{\overline{x}} \tag{1}$$

$$x_k^L \le x_k \le x_k^U, \ k = \overline{1, N} \tag{2}$$

$$g_i(\overline{x}) \le 0, \ i = \overline{1, t} \tag{3}$$

$$h_j(\overline{x}) = 0, \ j = \overline{1, p} \tag{4}$$

where $\overline{x}$ is a vector of objective variables; $f(\overline{x}) : R^N \to R$, $g(\overline{x})$ and $h(\overline{x})$ are the objective (fitness) function with $N$ objective variables, inequality constraints, and equality constraints, respectively; $t$ and $p$ are the numbers of inequality and equality constraints; and $x_k^L$ and $x_k^U$ are the lower and upper bounds for the $k$-th variable.

In concordance with the LSGO survey [2], there are two popular and effective ways of solving LSGO problems. The first group of methods are called non-decomposition methods. The second group of methods are cooperative coevolution (CC) methods. On the one hand, methods without decomposition use EA principles, which are especially developed for optimizing the whole high-dimensional vector [5,6]. On the other hand, CC methods [7,8] decompose the optimization vector into several pieces and evaluate them one by one using some Eas. The CC framework, together with EA as an optimizer, demonstrate a better performance in solving LSGO problems than many standard Eas and other optimization techniques. The main advantage of CC is that it is able to decrease the search space for applying an optimization algorithm, and thus it helps in dealing with the "curse of dimension" (CoD) [9].

In this study, an original improvement of CC for solving cLSGO tasks has been proposed and investigated. The main idea of the new approach is that it is permanently increasing the number of variables in subcomponents during an EA run. We have titled it "iCC". The SHADE (success-history based parameter adaptation for differential evolution) [10] was used as the main meta-heuristic (so-called "optimization core") in CC. The SHADE algorithm self-adapts some inner control parameters. These parameters are called "scale factor" (*F*) and "crossover rate" (*CR*).

In the last years, many extended EA techniques for handling constraints have been proposed, for example [11–14]. Additional details about constrained-handling approaches can be found in the following survey [15]. In this study, we have chosen the $\varepsilon$-DE technique [16], because it is well studied and popular. The whole proposed algorithm ($\varepsilon$-iCC-SHADE) combines the SHADE (optimizer), iCC (framework for solving large optimization problems), and $\varepsilon$-DE for the constraint handlings. In this paper, the performance of the $\varepsilon$-iCC-SHADE has been compared with the previously proposed $\varepsilon$-CC-SHADE with the different numbers of the population sizes and different numbers of subcomponents. The numerical experiments have demonstrated that the proposed $\varepsilon$-iCC-SHADE performs better than the $\varepsilon$-CC-SHADE.

In addition to our previous study [17], this paper was extended by new numerical experiments. The performance of the $\varepsilon$-iCC-SHADE algorithm has been investigated with different mutation strategies and different population sizes on the scaled problems from the IEEE CEC 2017 Competition on constrained real-parameter optimization. It is well-known that the EA performance strongly depends on fine-tuning its parameters. SHADE can only self-adapt two parameters (*F* and *CR*). We have chosen twelve well-known DE mutation strategies and have applied them to the $\varepsilon$-iCC-SHADE algorithm. All numerical experiments in this article are confirmed using Mann–Whitney U and Holm post-hoc tests.

The rest of the article has the following structure: In Section 2, we describe the research approaches; in Section 3, ε-iCC-SHADE is described; in Section 4, the experimental setup and results of numerical experiments are discussed; and in the Conclusion, the results and further research are discussed.

## 2. Related Work

### 2.1. Success-History Based Adaptive Differential Evolution (SHADE) Algorithm

Differential evolution (DE), first introduced by Storn and Price [18,19], is a powerful and efficient approach for solving BB optimization problems. Many DE-based algorithms have also proven their efficiency [20,21]. There are three main control parameters in the classic DE approach, they are *pop_size* (population size), *F* (scale factor), and *CR* (crossover rate). As we apply Eas for solving BB optimization problems, a random choice of parameter values is inappropriate. SHADE, proposed by Tanabe and Fukunaga [10], is one of the DE variants with self-adaptive mechanisms of *F* and *CR* parameters. The SHADE algorithm records the successful parameter values which improve fitness values during the run, and uses this knowledge to generate new values of *F* and *CR*. In addition, SHADE transfers replaced individuals into an external archive that maintains the previous experience and applies them at the mutation stage.

### 2.2. Cooperative Coevolution

To date, cooperative coevolution is one of the most effective frameworks for solving optimization problems with a large number of variables. CC was proposed by Potter and De Jong in [22,23]. Its basic idea was to divide a vector of the optimization task sequentially into pieces and apply some (meta)heuristic for solving these subproblems (this study used the SHADE algorithm as a core optimizer in CC). As a result of applying CC, the dimensionality and complexity of the LSGO problems decrease. The total number of subcomponents strongly influences the CC performance. It was noted earlier that LSGO tasks were viewed as BB problems. The functional relationship between variables is unknown. In this paper, CC performs with an equal size of subcomponents. It uses the following rule: $s \cdot m = N$, where $s$ is the number of variables in a subcomponent, $m$ is the total number of subcomponents, and $N$ is the total number of objective variables.

### 2.3. εDE Constrained Handlings

Many constrained-handlings techniques have been proposed [15]. In this study, we used the εDE approach [16] which transforms the selection operator in DE using Formula (5).

$$(f(X_1), v(X_1)) <_\varepsilon (f(X_2), v(X_2)) \leftrightarrow \begin{cases} f(X_1) < f(X_2) \; if \; v(X_1), v(X_2) \leq \varepsilon \\ f(X_1) < f(X_2) \; if \; v(X_1) = v(X_2) \\ v(X_1) < v(X_2), \; otherwise \end{cases} \tag{5}$$

where $f(X)$ is a fitness value of X solution and $v(X)$ is a value of constraints violation Formula (6):

$$v(X) = \frac{\sum_{i=1}^{p} G_i(X) + \sum_{j=1}^{k} H_j(X)}{p + k} \tag{6}$$

$$G_i(X) = \begin{cases} g_i(X), \; if \; g_i(X) > 0 \\ 0, \; otherwise \end{cases} \tag{7}$$

$$H_i(X) = \begin{cases} |h_j(X)|, \; if \; |h_j(X)| - \epsilon > 0 \\ 0, \; otherwise \end{cases} \tag{8}$$

In Formula (8), $\epsilon$ is the tolerance threshold, and it is equal to 0.0001 for all equality constraints. Analyzing Formula (5), we can conclude that the violation values have higher priority than the fitness

function values as comparing with the two-candidate solutions. To control the $\varepsilon$ parameter in Formula (5), we use the following modified Formula (9), inspired by [24].

$$\varepsilon = \begin{cases} E, \; if \; FEV \leq 0.8 \cdot MaxFEV \\ 0, \; otherwise \end{cases}$$
$$E = \left( \left( 1 - \frac{FEV}{MaxFEV} \right)^{c_p} \cdot v \left( X_{[\theta \cdot pop\_size]} \right) \right)$$

(9)

where *FEV* and *MaxFEV* are the current number of fitness evaluations and a maximum budget of fitness evaluations for the current run, respectively; $v\left( X_{[\theta \cdot pop\_size]} \right)$ is a violation value for solution $X$ with index $[\theta \cdot pop\_size]$ after sorting population (from best to worse); $c_p$ is a parameter for controlling the speed of constraints; $c_p$ is equal to 3. *Pop_size* is the population size; and $\theta$ is equal to 0.8.

Different values for $c_p$ and $\theta$ parameters have been tested. The following set of parameters has shown better performance: in Formula (9), $E$ is equal to 0 after evaluation of 80% of the total budget of FEV. After 80% of fitness evaluation in the last generations, EA should concentrate search in a local area.

## 3. Proposed Approach

EA demonstrates quite good performance in solving hard tasks of optimization, provided by the following heuristic rule. In the first generations, EA should use principles of exploration, i.e., perform a global search. Towards the end of the optimization process, EA should use methods of exploitation, i.e., perform a local search in some domain of the best-found solutions [25].

For decreasing the number of subcomponents in CC, we have used the following Scheme (10). Where $m$ is the number of subcomponents in CC. For BB problems, we do not know exactly how to allocate computational resources (the fitness budget). Thus, the proposed approach (iCC) has an equal amount of the objective function evaluations at each stage. We have also tried an alternative strategy for reducing the number of variables in each group, but this strategy has demonstrated low efficiency in solving cLSGO problems.

$$m = \begin{cases} 10, \; if \; FEV \epsilon [0, t_1] \\ 8, \; if \; FEV \epsilon [t_1 + 1, t_2] \\ 4, \; if \; FEV \epsilon [t_2 + 1, t_3] \\ 2, \; if \; FEV \epsilon [t_3 + 1, t_4] \\ 1, \; if \; FEV \epsilon [t_4 + 1, MaxFEV] \end{cases}$$
$$\forall \, i : \; t_{i+1} - t_i = 0.2 \cdot MaxFEV$$

(10)

At the stage of dividing, the LSGO problem into subproblems, an EA groups the variables into subcomponents of some predefined size. In the case of small groups, an EA usually demonstrates much faster improvement of the average fitness. At the same time, an EA performs a kind of local, coordinate-wise-like search when optimizing small groups and performs well only for appropriate combinations of variables in groups for separable and partially separable problems. For improving the performance of the LSGO EA, we proposed the following approach for tuning the grouping size: initialize CC with groups of small sizes and permanently increase sizes during the run of EA. Finally, in the last generations, we optimize the only component, which contains all variables. We distribute our fitness evaluation budget in an equal portion for all stages with different sizes in groups.

Table 1 demonstrates the pseudocode of the $\varepsilon$-iCC-SHADE. The termination criterion is the FEV budget exhaustion. It is also necessary to set two algorithm's parameters. These parameters are *MaxFEV* and *pop_size*. *MaxFEV* is the maximum number of fitness evaluations during an independent run and *pop_size* is the number of individuals in one CC subcomponent.

**Table 1.** $\varepsilon$-iCC-SHADE pseudocode.

| Line | Pseudocode |
|------|-----------|
| 1: | Set *MaxFEV*, *pop_size* |
| 2: | **while** (termination criterion is not satisfied) **do** |
| 3: | Calculate m = Equation (10); |
| 4: | **while** (i < m +1) **do** |
| 5: | Divide optimization vector into m subcomponents (); |
| 6: | i = 1; |
| 7: | Evaluate the i-th subcomponent with $\varepsilon$-SHADE$_i$ algorithm (); |
| 8: | i = i + 1; |
| 9: | end **while** |
| 10: | end **while** |
| 11: | Return best-found value and finish optimization procedure (); |

## 4. Numerical Experiments and Results

### 4.1. Benchmark Set for Constrained Large-Scale Global Optimization Problems

Today, there is no special cLSGO benchmark set to evaluate optimization algorithms. A new benchmark has been proposed based on scalable problems from the IEEE CEC 2017 Competition on constrained single objective real-parameter optimization (CSORPO) [26]. We selected and modified those problems from the IEEE CEC 2017 benchmark which did not use high-dimensional transformation matrixes (these matrixes have been manually designed for the specific number of variables only). The selected problems were scaled up to one thousand variables and were included in the proposed set of cLSGO problems. Table 2 contains information about the enumeration of functions that are included in the cLSGO benchmark and about types of objective functions. Here, N and S define non-separable and separable problems, respectively.

**Table 2.** Scaled optimization problems for the constrained large-scale global optimization (cLSGO) benchmark set.

| cLSGO Problem | 1 | 2 | 3 | 4 | 5 | 6 | 7 | 8 | 9 | 10 | 11 | 12 | 13 | 14 | 15 | 16 | 17 | 18 |
|---------------|---|---|---|---|---|---|---|---|---|----|----|----|----|----|----|----|----|----|
| CEC'2017 CSORPO | 1 | 3 | 4 | 6 | 7 | 8 | 9 | 10 | 11 | 12 | 13 | 14 | 15 | 16 | 17 | 18 | 19 | 20 |
| Objective Type | N | N | S | S | S | S | S | S | S | S | N | N | S | S | N | S | S | N |

### 4.2. Software Implementation and Setups, Benchmark Set for Constrained Large-Scale Global Optimization Problems

To evaluate our algorithms, we used the following computational system. The operating system was Ubuntu Linux 18.04 LTS. To decrease computation time, we used two CPUs with 32 threads (Ryzen 7 1700x (8C/16T) and Ryzen 2700 (8C/16T)). The software was developed using the C++ language (g++(gcc) compiler) and Code:Blocks 17.12.

To investigate the performance of each algorithm correctly, it is necessary to ensure equal experimental conditions. The dimension for all benchmark problems is $D = 1000$. The number of independent runs is 25 per benchmark problem. *MaxFEV* is equal to 3E+6 for an independent run for each algorithm and for each problem.

### 4.3. Investigation of the Performance of iCC Framework versus Classic Cooperative Coevolution with the Fixed Number of Subcomponents

We use the following notation "$\varepsilon$-CC-SHADE (m)", where *m* is the total fixed number of subcomponents that the algorithm uses in the optimization procedure. We investigate $\varepsilon$-CC-SHADE (m) with different numbers of subcomponents (1, 2, 4, 8, and 10) and different numbers of population sizes (25, 50, 75, and 100). In addition, if *m* is equal to 1, then the algorithm optimizes a problem without the CC framework, because it uses only one subcomponent, which has a size equal to the optimization

vector. The ε-iCC-SHADE was investigated with the different numbers of the population size (25, 50, 75, and 100). The rule of increasing the total number of variables in groups is described in Equation (10). It is worth mentioning that, to evaluate some generation with *m* subcomponents and *pop_size* number of individuals, it is required to calculate *m·pop_size* solutions. Hence, on the one hand, if we reduce the number of subcomponents, the algorithm needs to evaluate fewer solutions in each generation. Thus, the total number of generations increase, provided that *MaxFEV* is fixed. With regards to the iCC framework, at the end of the optimization process, the algorithm can calculate many generations, as a consequence, it finds better local solutions. As a result, the performance increases. On the other hand, the strategy of increasing the total number of subcomponents has demonstrated bad performance.

Figure 1 demonstrates the average ranking results for the Eas performance with population size equal to 25, 50, 75, and 100, respectively, using boxplot diagrams. On the x-axis, there are labels of optimization algorithms. On the y-axis, the average ranks for each algorithm are presented. The best EA has the smallest average rank (median value). The ranking is based on the median best-found values averaged over all cLSGO benchmark problems. Figures 2 and 3 demonstrate the average ranking results for non-separable and separable problems, respectively. Figures 2 and 3 have the same structure as Figure 1. As we can see from Figure 1, the proposed ε-iCC-SHADE performs better on average than ε-CC-SHADE (m) with different fixed numbers of subcomponents. Figure 2 shows that ε-iCC-SHADE does not demonstrate a better average performance on non-separable problems. At the same time, Figure 3 shows that ε-iCC-SHADE outperforms ε-CC-SHADE (m) by the average rank on separable problems.

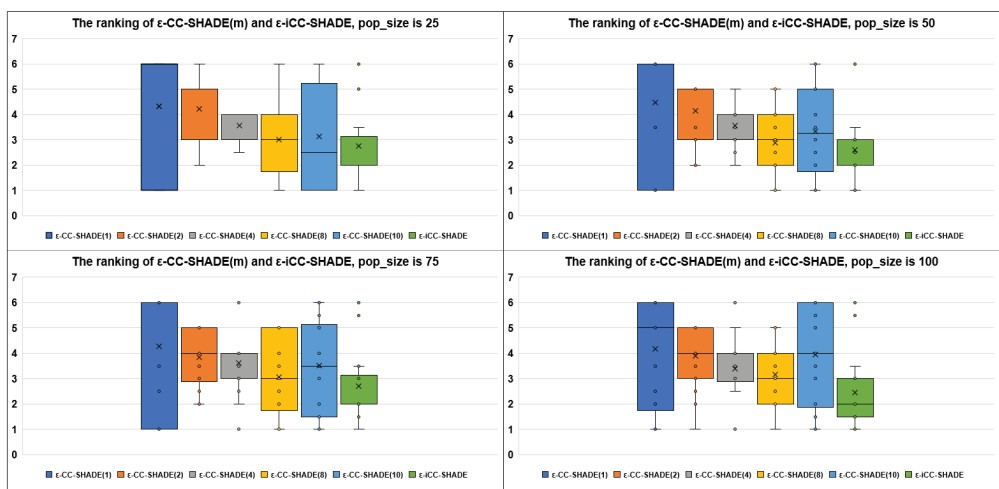

**Figure 1.** The ranking of ε-CC-SHADE (m) and ε-iCC-SHADE on all problems.

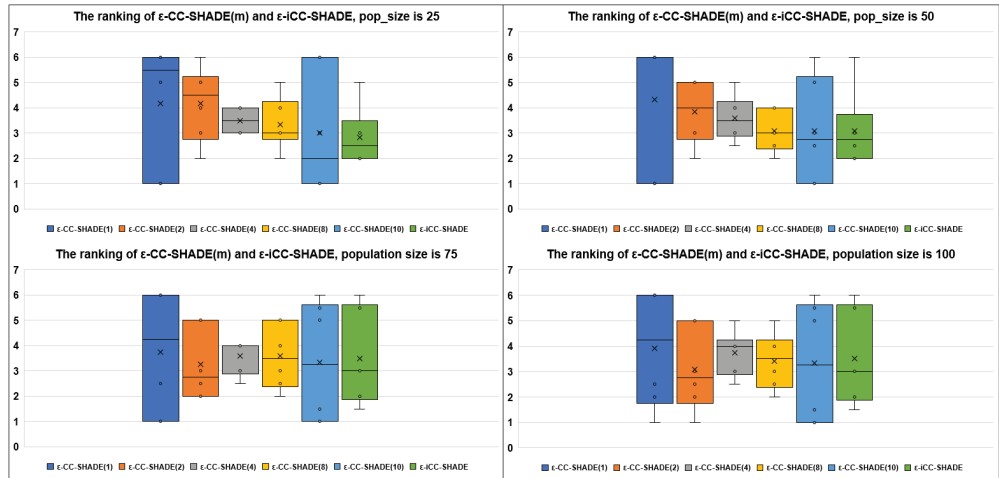

**Figure 2.** The ranking of ε-CC-SHADE (m) and ε-iCC-SHADE on non-separable problems.

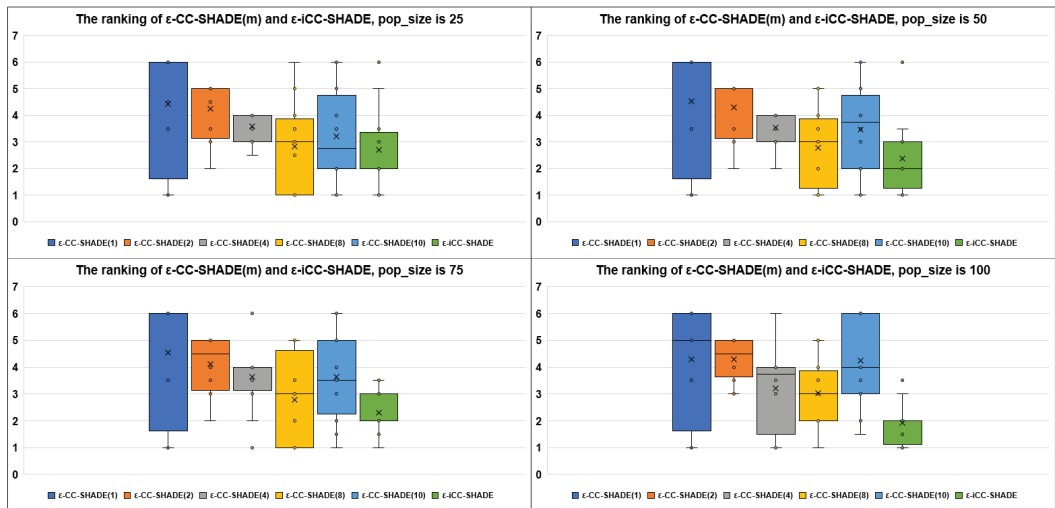

**Figure 3.** The ranking of $\varepsilon$-CC-SHADE (m) and $\varepsilon$-iCC-SHADE on separable problems.

We have applied the following sorting method for solutions in populations, in accordance with the benchmark rules [26]: feasible solutions are always better than an infeasible solution and all feasible solutions are sorted based on fitness function values. The results of algorithms which rank the results for each cLSGO problem are presented in Figures 4–11 (the lowest rank corresponds to the best algorithm) using parallel diagram plots. On the x-axis, there are labels for the problem number; on the y-axis, there are ranks for each algorithm.

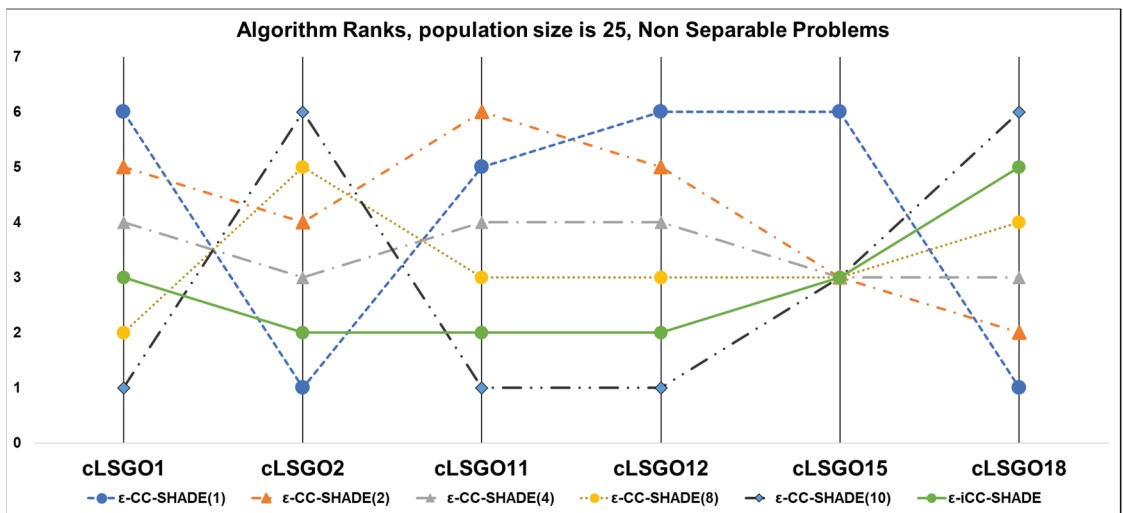

**Figure 4.** The ranking of $\varepsilon$-CC-SHADE (m) and $\varepsilon$-iCC-SHADE for non-separable problems.

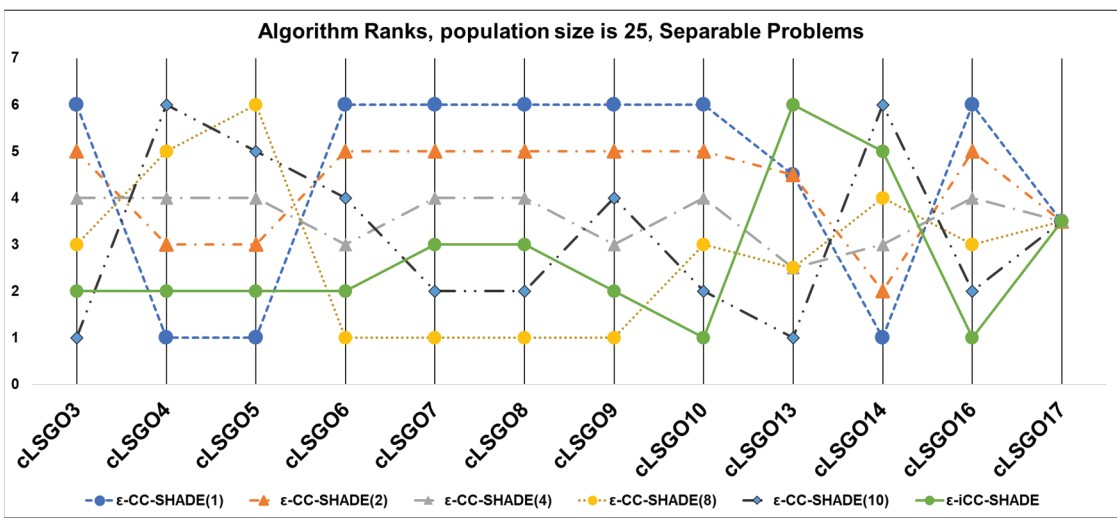

**Figure 5.** The ranking of $\varepsilon$-CC-SHADE (m) and $\varepsilon$-iCC-SHADE for separable problems.

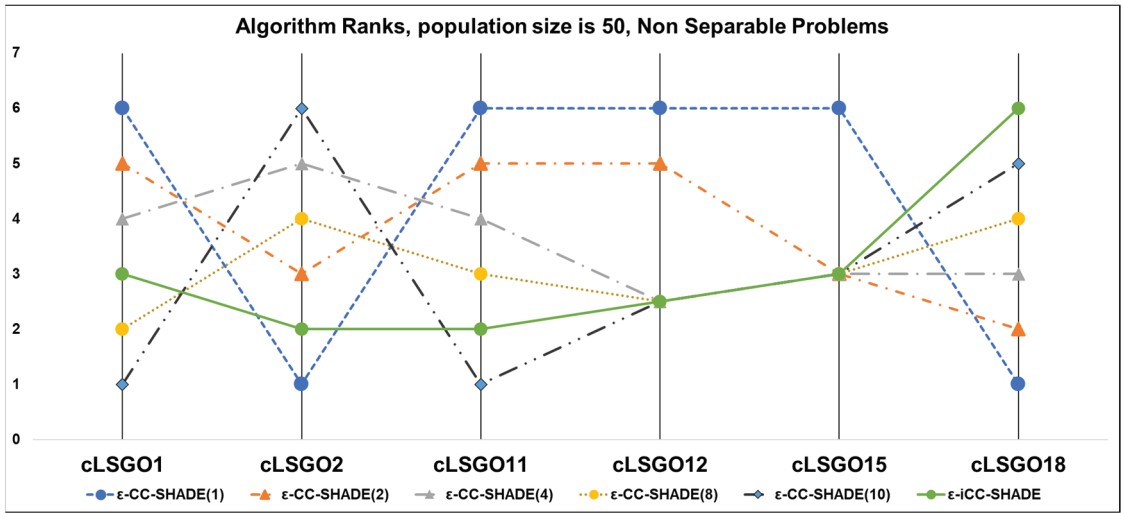

**Figure 6.** The ranking of $\varepsilon$-CC-SHADE (m) and $\varepsilon$-iCC-SHADE for non-separable problems.

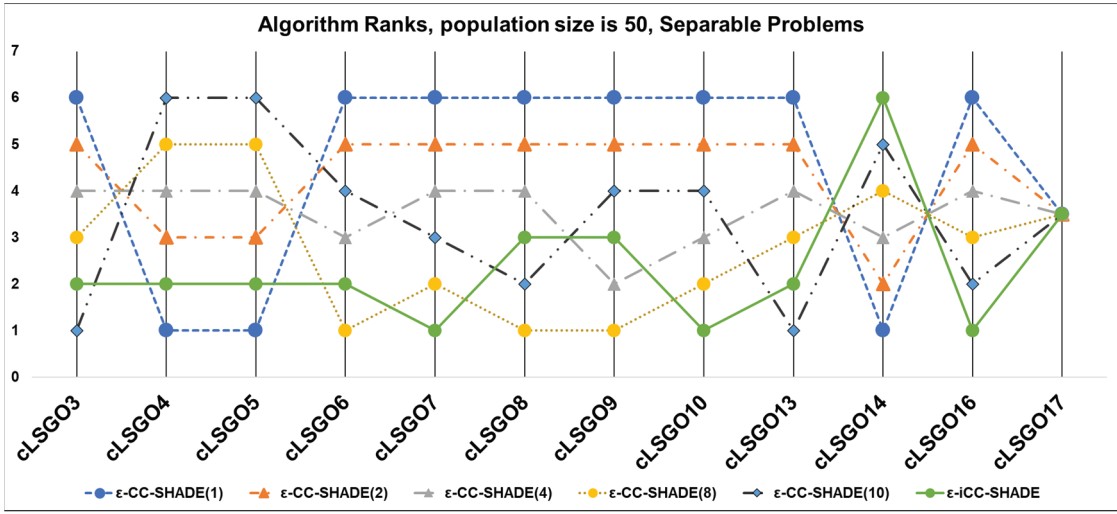

**Figure 7.** The ranking of $\varepsilon$-CC-SHADE (m) and $\varepsilon$-iCC-SHADE for separable problems.

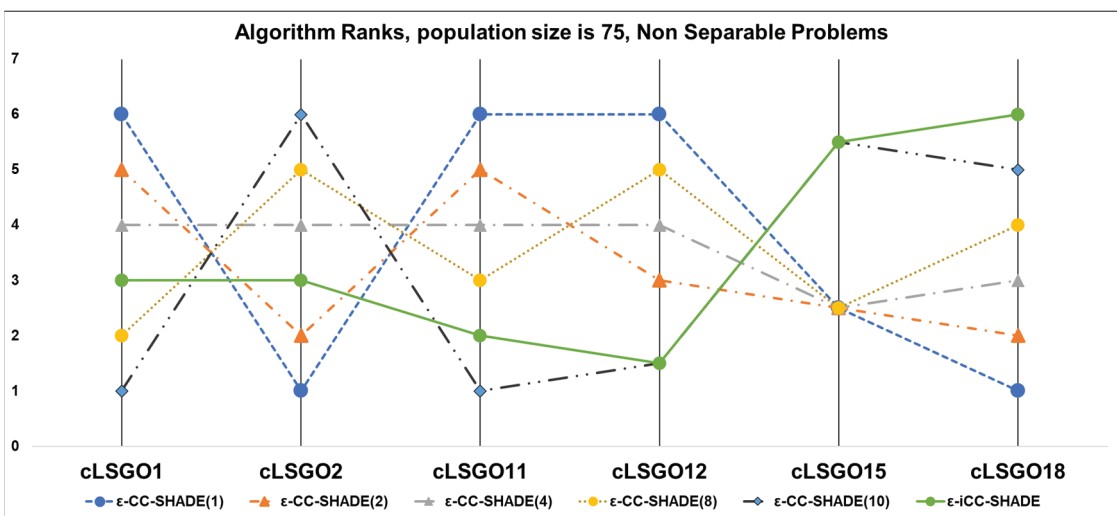

**Figure 8.** The ranking of ε-CC-SHADE (m) and ε-iCC-SHADE for non-separable problems.

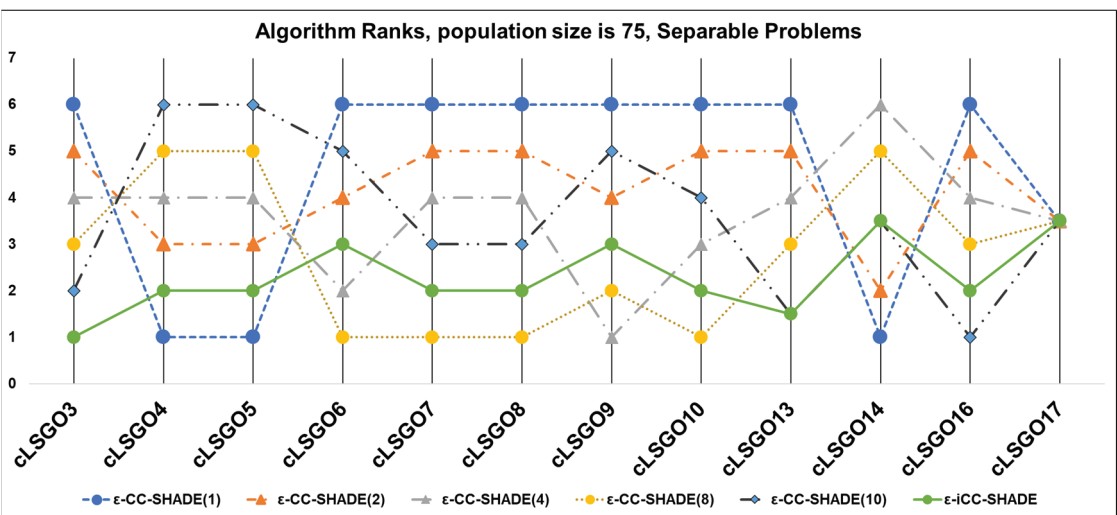

**Figure 9.** The ranking of ε-CC-SHADE (m) and ε-iCC-SHADE for separable problems.

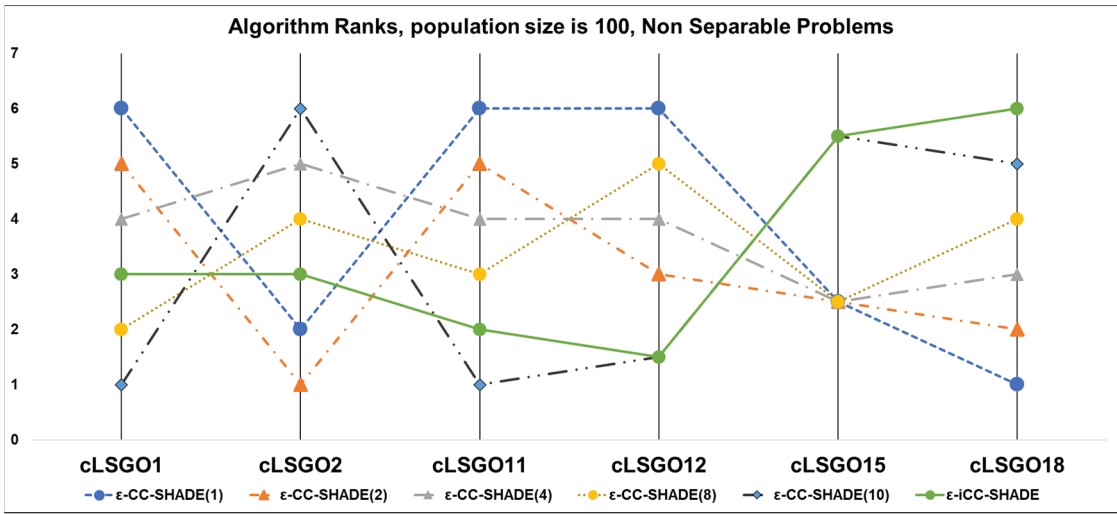

**Figure 10.** The ranking of ε-CC-SHADE (m) and ε-iCC-SHADE for non-separable problems.

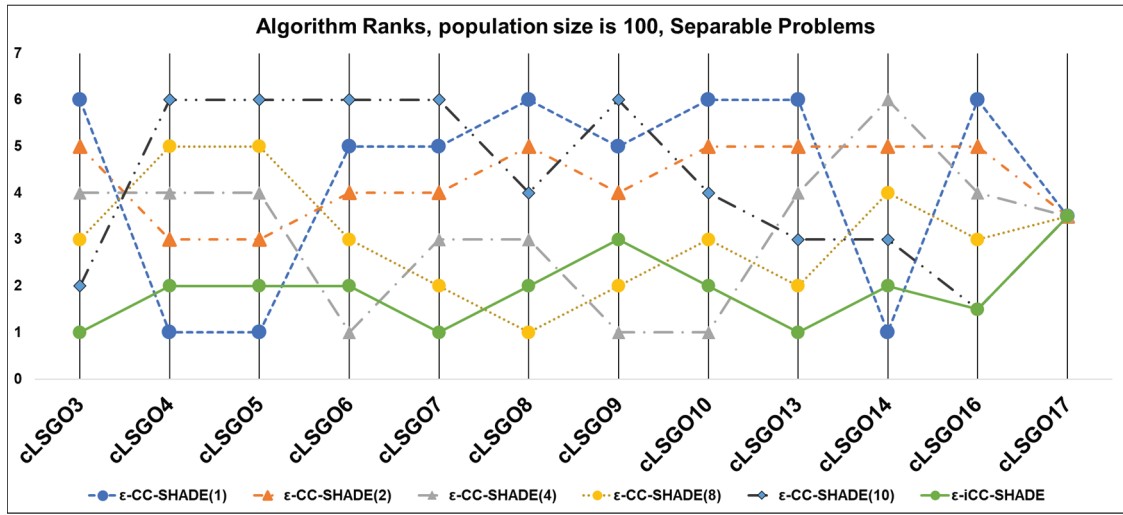

**Figure 11.** The ranking of $\varepsilon$-CC-SHADE (m) and $\varepsilon$-iCC-SHADE for separable problems.

Tables 3–6 prove the statistical difference in the results of estimating the performance of $\varepsilon$-iCC-SHADE (iCC) and $\varepsilon$-CC-SHADE (m) (CC (m)) using the Mann–Whitney U test with normal approximation and tie correction with p-value equal to 0.01. The first column contains the names of the algorithms, the second, third, and fourth columns contain the numbers of benchmark problems, and the EA in the head of the table shows better (+), worse (−), and equal performance ($\approx$). The performance of the proposed $\varepsilon$-iCC-SHADE versus $\varepsilon$-CC-SHADE (m) algorithms has been investigated. Tables 3–6 demonstrate that $\varepsilon$-iCC-SHADE is statistically better than $\varepsilon$-CC-SHADE (m).

**Table 3.** Mann–Whitney U test, pop_size is 25.

| iCC vs. | + | − | $\approx$ |
|---|---|---|---|
| CC (1) | 11 | 3 | 4 |
| CC (2) | 14 | 2 | 2 |
| CC (4) | 13 | 2 | 3 |
| CC (8) | 7 | 3 | 8 |
| CC (10) | 5 | 2 | 11 |
| Total amount | 50 | 12 | 28 |

**Table 4.** Mann–Whitney U test, pop_size is 50.

| iCC vs. | + | − | $\approx$ |
|---|---|---|---|
| CC (1) | 13 | 4 | 1 |
| CC (2) | 14 | 2 | 2 |
| CC (4) | 13 | 2 | 3 |
| CC (8) | 5 | 2 | 11 |
| CC (10) | 6 | 2 | 10 |
| Total amount | 51 | 12 | 27 |

**Table 5.** Mann–Whitney U test, pop_size is 75.

| iCC vs. | + | − | $\approx$ |
|---|---|---|---|
| CC (1) | 12 | 5 | 1 |
| CC (2) | 14 | 3 | 1 |
| CC (4) | 11 | 2 | 5 |
| CC (8) | 6 | 3 | 9 |
| CC (10) | 7 | 2 | 9 |
| Total amount | 50 | 15 | 25 |

**Table 6.** Mann–Whitney U test, pop_size is 100.

| iCC vs. | + | − | ≈ |
|---|---|---|---|
| CC (1) | 12 | 4 | 2 |
| CC (2) | 15 | 2 | 1 |
| CC (4) | 11 | 2 | 5 |
| CC (8) | 9 | 3 | 6 |
| CC (10) | 7 | 3 | 8 |
| Total amount | 54 | 14 | 22 |

In addition to the Mann–Whitney U test, we used post-hoc Dunn and Kruskal–Wallis multiple comparison p-values adjusted with the Holm post-hoc test; p-value equals 0.05. The comparison results are presented in Table 7. The values, in Table 7, are based on the number of cases when the performance estimates were significantly different. Each value indicates the sum of performance differences for all cLSGO problems. The table rows indicate the number of individuals. The last row shows the total difference sum. As we can see from Table 7, $\varepsilon$-CC-SHADE (4) and $\varepsilon$-CC-SHADE (8) demonstrate the minimum value for the sum of differences. At the same time, $\varepsilon$-CC-SHADE (1) demonstrates the biggest performance differences versus other algorithms. Both $\varepsilon$-CC-SHADE (10) and the proposed $\varepsilon$-iCC-SHADE demonstrate medium values of differences.

**Table 7.** The Holm test, $\varepsilon$-CC-SHADE (m) vs. $\varepsilon$-iCC-SHADE p-value = 0.05.

| Population Size | CC (1) | CC (2) | CC (4) | CC (8) | CC (10) | iCC |
|---|---|---|---|---|---|---|
| 25 | 65 | 53 | 52 | 49 | 51 | 52 |
| 50 | 72 | 55 | 48 | 46 | 54 | 53 |
| 75 | 74 | 56 | 55 | 52 | 56 | 55 |
| 100 | 69 | 55 | 48 | 48 | 54 | 54 |
| Total amount | 280 | 219 | 203 | 195 | 215 | 214 |

### 4.4. Effect of Mutation Strategy on the iCC Framework Performance

It is well known that there are many different mutation strategies for DE algorithms. In this subsection, we investigate the iCC performance with various mutation operators. Many operators have been taken from [27]. The detailed description of each mutation strategy can be found in Table 8. The first column of Table 8 contains the original names of mutation schemes, the second column contains the formula for evaluating mutant vectors, and the third column contains the short notations. We have modified the traditional mutation operators (mut-1, mut-2, mut-3, mut-4, mut-5, and mut-6) by applying the tournament selection for choosing indexes. The tournament size has been set to 2.

**Table 8.** Mutation strategies.

| Mutation Strategy | Formula | Reassignment |
|---|---|---|
| DE/rand/1 | $x_{r_1} + F \cdot (x_{r_2} - x_{r_3})$ | mut-1 |
| DE/rand/2 | $x_{r_1} + F \cdot (x_{r_2} - x_{r_3}) + F \cdot (x_{r_4} - x_{r_5})$ | mut-2 |
| DE/best/1 | $x_{best} + F \cdot (x_{r_2} - x_{r_3})$ | mut-3 |
| DE/best/2 | $x_{best} + F \cdot (x_{r_2} - x_{r_3}) + F \cdot (x_{r_4} - x_{r_5})$ | mut-4 |
| DE/cut-to-best/1 | $x_i + F \cdot (x_{best} - x_i) + F \cdot (x_{r_2} - x_{r_3})$ | mut-5 |
| DE/cur-to-*p*best/1 | $x_i + F \cdot \left(x_{r_{pbest}} - x_i\right) + F \cdot (x_{r_2} - x_{r_3})$ | mut-6 |
| DE/tour/1 | $x_{t_1} + F \cdot (x_{t_2} - x_{t_3})$ | mut-7 |
| DE/tour/2 | $x_{t_1} + F \cdot (x_{t_2} - x_{t_3}) + F \cdot (x_{t_4} - x_{t_5})$ | mut-8 |
| DE/best/1(tour) | $x_{best} + F \cdot (x_{t_2} - x_{t_3})$ | mut-9 |
| DE/best/2(tour) | $x_{best} + F \cdot (x_{t_2} - x_{t_3}) + F \cdot (x_{t_4} - x_{t_5})$ | mut-10 |
| DE/cut-to-best/1(tour) | $x_i + F \cdot (x_{best} - x_i) + F \cdot (x_{t_2} - x_{t_3})$ | mut-11 |
| DE/cur-to-*p*best/1(tour) | $x_i + F \cdot \left(x_{t_{pbest}} - x_i\right) + F \cdot (x_{t_2} - x_{t_3})$ | mut-12 |

Figures 12 and 13 show boxplot diagrams for the results of ranking algorithms with different mutations for the population size 50 and 100, respectively. Figure 12 shows the average ranking results for the EAs performance with population size equal to 50 for all non-separable and separable cLSGO problems using boxplot diagrams. As we can see from Figure 12, on the one hand, mut-6, mut-7, and mut-12 perform better on all cLSGO benchmark problems, including separable problems. On the other hand, mut-6, mut-11, and mut-12 show better performance in solving non-separable problems.

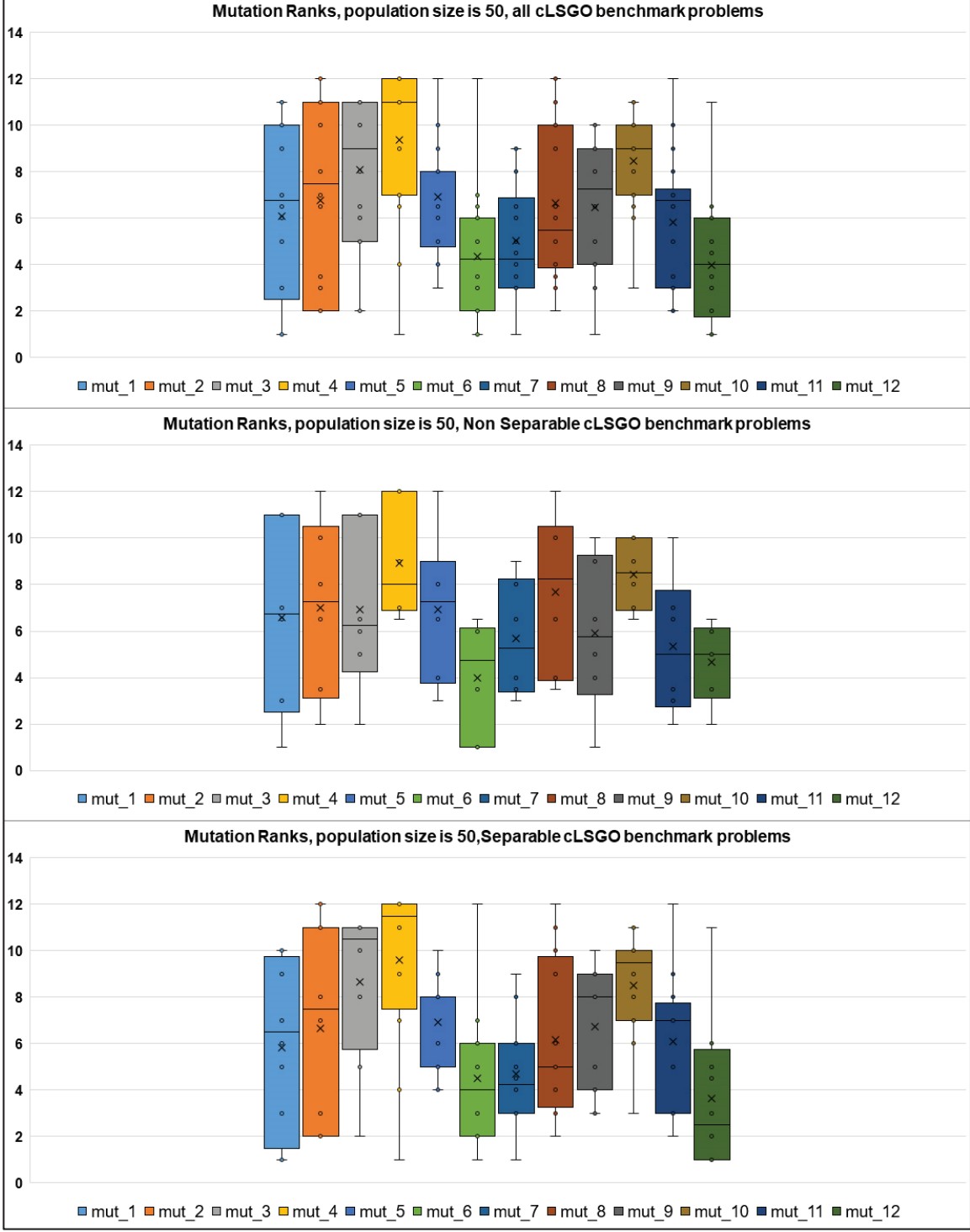

**Figure 12.** The ranking of different mutation strategies on the cLSGO benchmark problems.

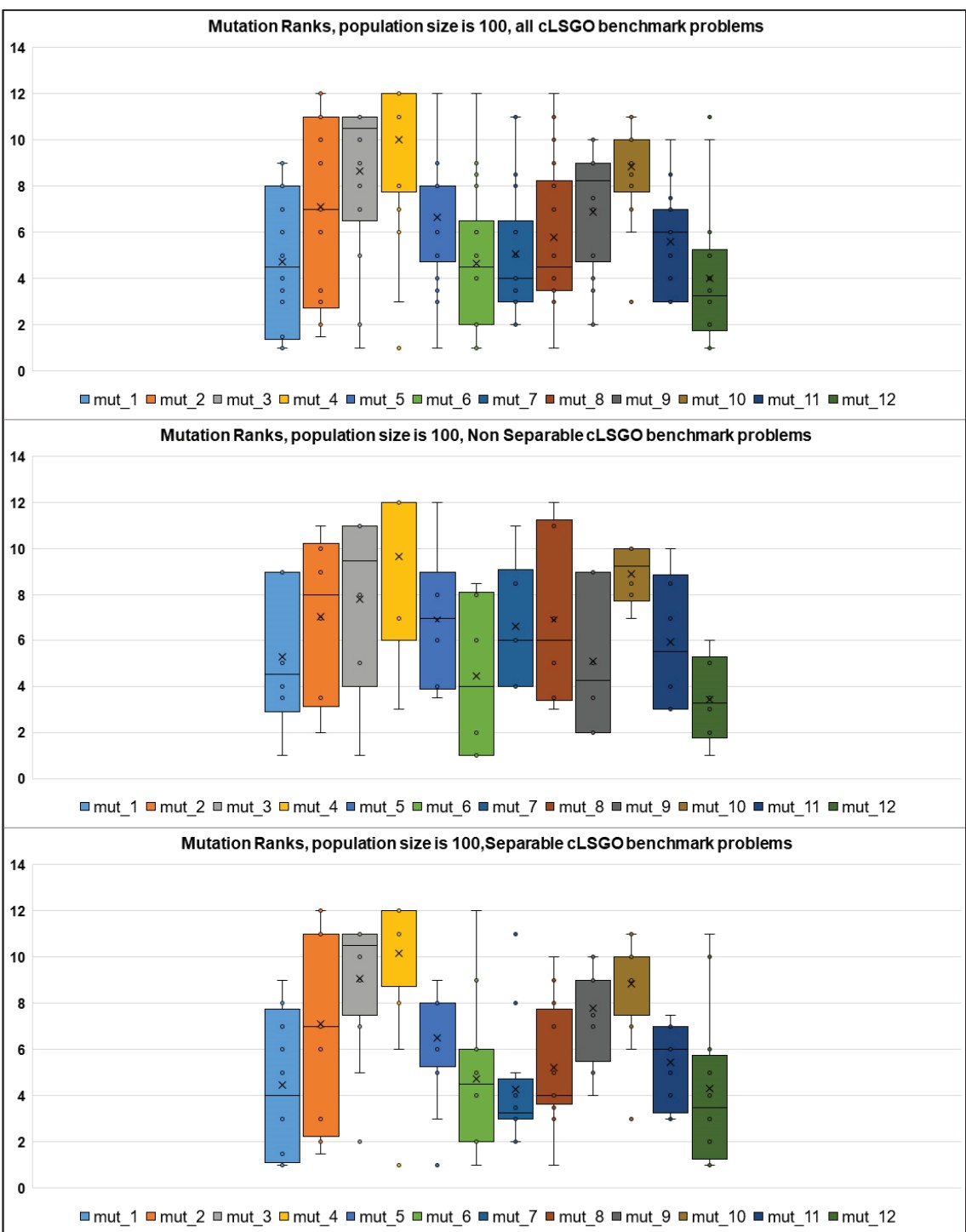

**Figure 13.** The ranking of different mutation strategies on the cLSGO benchmark problems.

As we can see from Figure 13, mut-1, mut-6, mut-7, and mut-12 perform better on all cLSGO benchmark problems, including separable problems, and mut-6, mut-9, and mut-12 show better performance on the non-separable cLSGO benchmark problems.

The mutation ranking results for each cLSGO problem are presented in Figures 14–17 (the lowest rank corresponds to the best algorithm) using parallel diagram plots. On the x-axis, we have problem numbers; on the y-axis, we have the rank for each mutation.

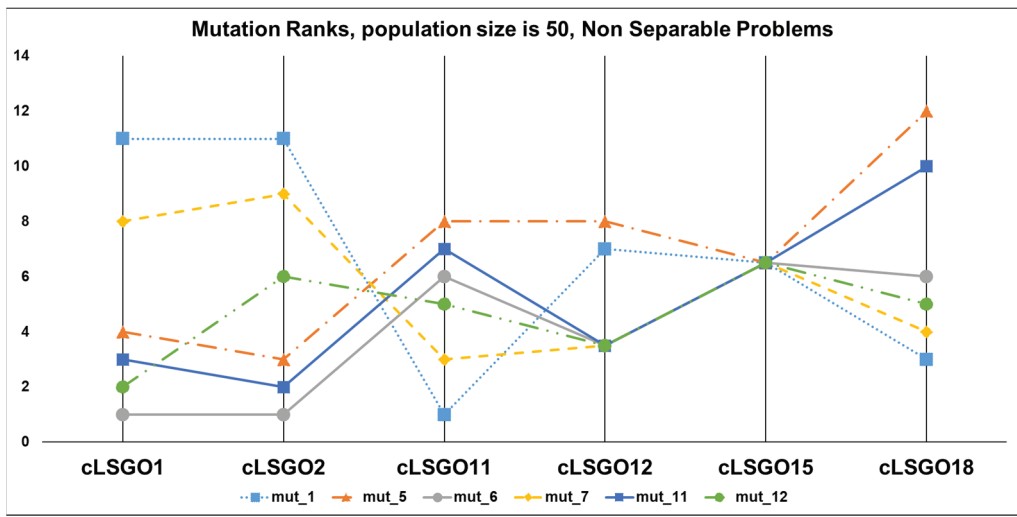

**Figure 14.** The ranking of ε-iCC-SHADE with some mutation strategies for non-separable problems.

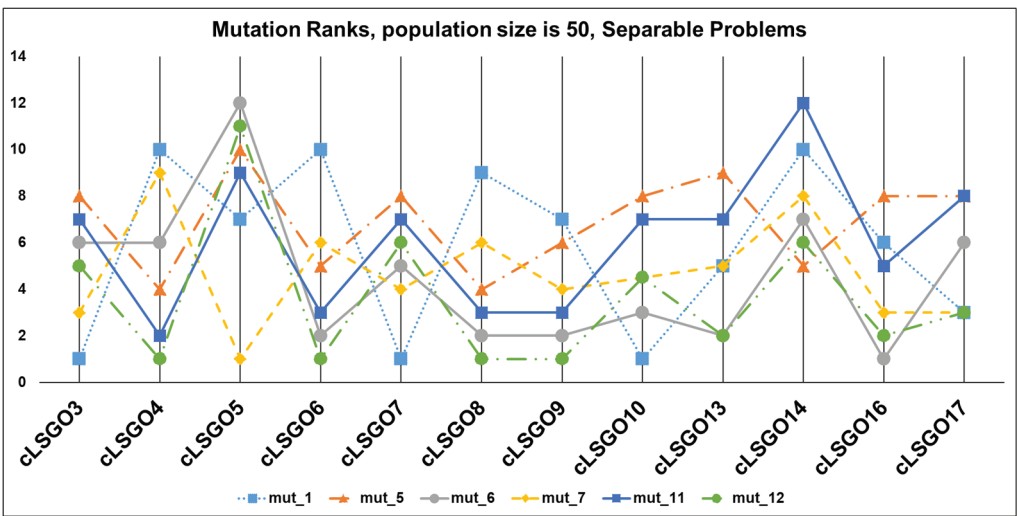

**Figure 15.** The ranking of ε-iCC-SHADE with some mutation strategies for separable problems.

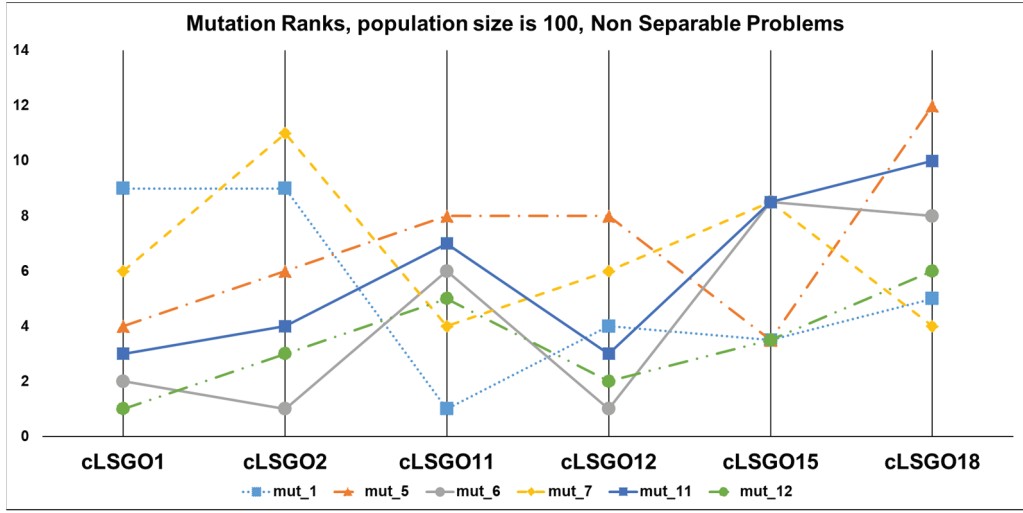

**Figure 16.** The ranking of ε-iCC-SHADE with some mutation strategies for non-separable problems.

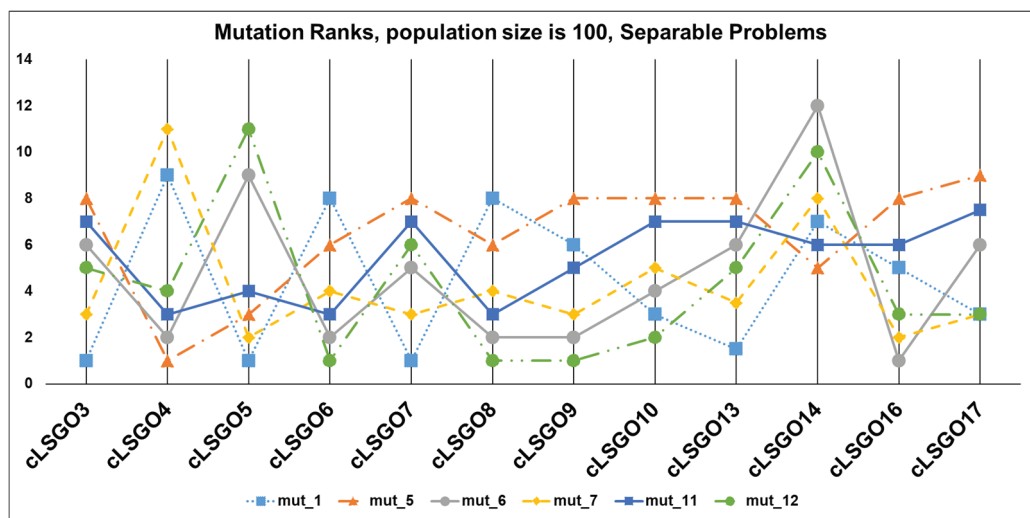

**Figure 17.** The ranking of $\varepsilon$-iCC-SHADE with some mutation strategies for separable problems.

Tables 9 and 10 show the results of the Mann–Whitney U test with normal approximation and tie correction with p = 0.01 for the pop_size equal to 50 and 100, respectively. In the tables, each cell contains the value, which has been calculated using the following algorithm. For each benchmark problem, if the mutation scheme from the corresponding column outperforms the mutation from the corresponding row, we add +1 to the score in the cell at the column-row crossing; otherwise, we add -1, and we add 0 for equal performances. The last column contains summary scores for all mutation schemes. The highest summary score corresponds to the best scheme.

**Table 9.** Results of the Mann–Whitney U test, the population size is 50.

|    | 1   | 2   | 3   | 4   | 5   | 6   | 7   | 8   | 9   | 10  | 11  | 12  | Total Sum |
|----|-----|-----|-----|-----|-----|-----|-----|-----|-----|-----|-----|-----|-----------|
| 1  | 0   | 6   | 11  | 14  | 9   | −1  | 2   | 8   | 8   | 11  | 6   | 2   | 76   |
| 2  | −6  | 0   | 8   | 9   | 4   | −4  | −4  | −1  | 6   | 9   | 2   | −3  | 20   |
| 3  | −11 | −8  | 0   | −5  | −13 | −13 | −12 | −10 | −8  | −5  | −12 | −11 | −108 |
| 4  | −14 | −9  | 5   | 0   | −12 | −13 | −13 | −11 | −8  | −6  | −12 | −11 | −104 |
| 5  | −9  | −4  | 13  | 12  | 0   | −11 | −9  | −6  | 8   | 10  | −8  | −10 | −14  |
| 6  | 1   | 4   | 13  | 13  | 11  | 0   | 1   | 4   | 11  | 13  | 11  | 0   | 82   |
| 7  | −2  | 4   | 12  | 13  | 9   | −1  | 0   | 5   | 9   | 12  | 9   | 2   | 72   |
| 8  | −8  | 1   | 10  | 11  | 6   | −4  | −5  | 0   | 8   | 11  | 2   | 0   | 32   |
| 9  | −8  | −6  | 8   | 8   | −8  | −11 | −9  | −8  | 0   | 4   | −11 | −10 | −51  |
| 10 | −11 | −9  | 5   | 6   | −10 | −13 | −12 | −11 | −4  | 0   | −11 | −10 | −80  |
| 11 | −6  | −2  | 12  | 12  | 8   | −11 | −9  | −2  | 11  | 11  | 0   | −10 | 14   |
| 12 | −2  | 3   | 11  | 11  | 10  | 0   | −2  | 0   | 10  | 10  | 10  | 0   | 61   |

**Table 10.** Results of the Mann–Whitney U test, the population size is 100.

|    | 1   | 2   | 3   | 4   | 5   | 6   | 7   | 8   | 9   | 10  | 11  | 12  | Total Sum |
|----|-----|-----|-----|-----|-----|-----|-----|-----|-----|-----|-----|-----|-----------|
| 1  | 0   | 7   | 8   | 9   | 6   | −4  | 0   | 6   | 6   | 9   | 6   | −2  | 51  |
| 2  | −7  | 0   | 7   | 8   | 3   | −3  | −5  | −1  | 6   | 9   | 2   | −3  | 16  |
| 3  | −8  | −7  | 0   | 5   | −10 | −10 | −9  | −8  | −7  | −4  | −9  | −9  | −76 |
| 4  | −9  | −8  | −5  | 0   | −8  | −10 | −11 | −9  | −9  | −6  | −11 | −9  | −95 |
| 5  | −6  | −3  | 10  | 8   | 0   | −14 | −7  | −5  | 7   | 10  | −7  | −13 | −20 |
| 6  | 4   | 3   | 10  | 10  | 14  | 0   | 3   | 3   | 9   | 12  | 15  | −1  | 82  |
| 7  | 0   | 5   | 9   | 11  | 7   | −3  | 0   | 6   | 8   | 12  | 8   | −2  | 61  |
| 8  | −6  | 1   | 8   | 9   | 5   | −3  | −6  | 0   | 7   | 9   | 6   | −3  | 27  |
| 9  | −6  | −6  | 7   | 9   | −7  | −9  | −8  | −7  | 0   | 7   | −9  | −9  | −38 |
| 10 | −9  | −9  | 4   | 6   | −10 | −12 | −12 | −9  | −7  | 0   | −11 | −10 | −79 |
| 11 | −6  | −2  | 9   | 11  | 7   | −15 | −8  | −6  | 9   | 11  | 0   | −13 | −3  |
| 12 | 2   | 3   | 9   | 9   | 13  | 1   | 2   | 3   | 9   | 10  | 13  | 0   | 74  |

Table 11 shows the results of the Holm post-hoc test (as Table 7). The first row of Table 11 presents the type of mutation schemes from 1 to 12. The second and the third rows present the total difference sum for 50 and 100 individuals, respectively. From the results of applying the Holm test, we can conclude that mutations under numbers 2, 4, 6, and 12 have the smallest performance difference. However, mutation under the numbers 5, 8, and 10, have the largest performance difference.

**Table 11.** Holm test, p-value = 0.05.

| Population Size | 1 | 2 | 3 | 4 | 5 | 6 | 7 | 8 | 9 | 10 | 11 | 12 |
|---|---|---|---|---|---|---|---|---|---|---|---|---|
| 50 | 58 | 55 | 64 | 50 | 68 | 54 | 54 | 65 | 61 | 63 | 59 | 53 |
| 100 | 58 | 53 | 62 | 53 | 63 | 49 | 67 | 61 | 64 | 65 | 58 | 51 |
| Total sum | 116 | 108 | 126 | 103 | 131 | 103 | 121 | 126 | 125 | 128 | 117 | 104 |

Convergence plots for the average fitness value and the average violation value for benchmark Problem 6 and *pop_size* is 50 are presented in Figure 18. As we can see, violation values are always decreasing or remain at the same level. At the same time, the $\varepsilon$ constrained handling (Equation (5)) can lead to increasing fitness values. We can see the same behavior of algorithms in Figure 19.

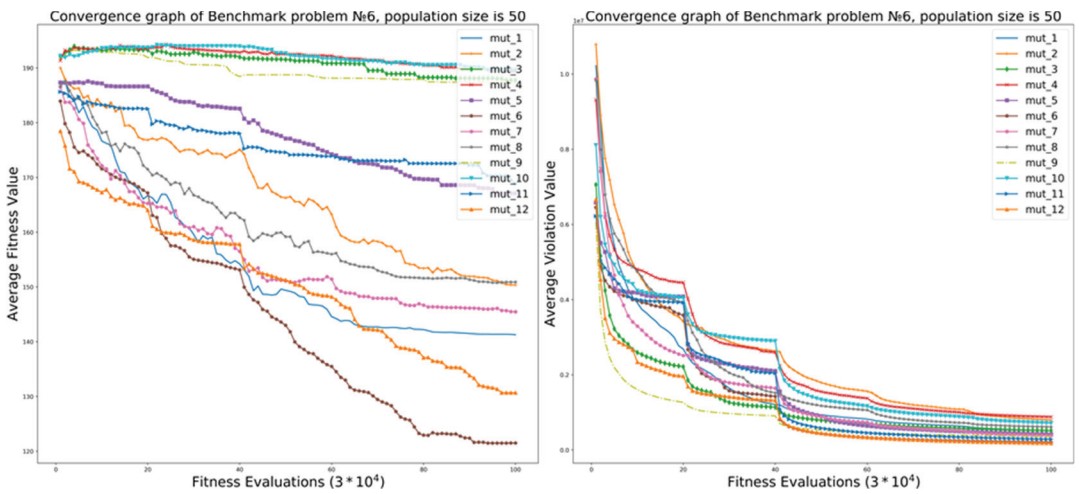

**Figure 18.** The performance of iCC framework on cLSGO benchmark Problem 6.

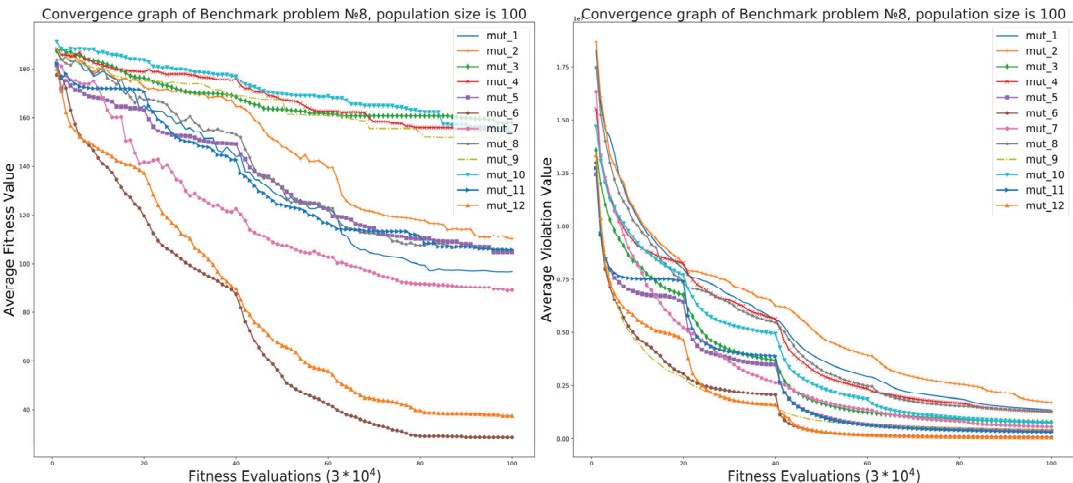

**Figure 19.** The performance of iCC framework on cLSGO benchmark Problem 8.

As we can see in Figures 18 and 19, the convergence lines of violations are increasing the convergence speed every 20% of FEVs, when iCC changes the number of variables in subcomponents.

## 5. Conclusions

In this paper, a novel ε-iCC-SHADE was proposed for solving constrained large-scale global optimization problems. The ε-iCC-SHADE is based on iCC. The iCC method uses the strategy of increasing the number of variables in subcomponents. The performance of ε-iCC-SHADE was investigated with different population sizes. The ε-iCC-SHADE performance was compared with the early proposed ε-CC-SHADE. The numerical experiments showed that the improved iCC method demonstrated better results than the classic CC with a constant number of subcomponents. The ε-iCC-SHADE outperformed, on average, all ε-CC-SHADE variants when ranking using all cLSGO benchmark problems and only separable problems. The ε-iCC-SHADE had a lower number of control parameters and we did not need to tune the number of subcomponents. Hence, we only needed to set the population size to run ε-iCC-SHADE. Additionally, we investigated the iCC performance with twelve mutation strategies and compared their performances. On the basis of the results of numerical experiments, we conclude that DE/rand/1, DE/cur-to-pbest/1, DE/tour/1, and DE/cur-to-pbest/1(tour) mutation strategies outperform all others. These mutation strategies perform better, on average, for all cLSGO benchmark problems. As we can see from the results, iCC has a high potential for enhancement. In further studies, we would try to design an adaptive scheme for changing group sizes.

**Author Contributions:** Conceptualization, A.V. and E.S.; methodology, A.V. and E.S; software, A.V.; validation, A.V. and E.S.; formal analysis, A.V. and E.S.; investigation, A.V. and E.S.; resources, A.V. and E.S.; data curation, A.V. and E.S.; writing—original draft preparation, A.V. and E.S.; writing—review and editing, A.V. and E.S.; visualization, A.V. and E.S.; supervision, E.A.; project administration, A.V. and E.S; funding acquisition, A.V. and E.S. All authors have read and agreed to the published version of the manuscript.

**Funding:** This research received no external funding.

**Conflicts of Interest:** The authors declare no conflict of interest.

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
