# Peer review of "Investigation of the iCC Framework Performance for Solving Constrained LSGO Problems†"

_algorithms, doi:10.3390/a13050108_

Round 1
Reviewer 1 Report
It is an interesting and timely study. I would recommend adding a subsection on the failure modes of the proposed algorithm will concretize the work better. Also, the manuscript is incomplete in the sense that figure 12 cannot be visualized - please address this issue and resubmit for re-evaluation.
Author Response
First, we would like to thanks the reviewer for so detailed and very useful remarks.
It is an interesting and timely study. I would recommend adding a subsection on the failure modes of the proposed algorithm will concretize the work better.
This information can be obtained from out experimental results. Algorithms with low ranks are exactly such “the failure modes”. We also have presented statistical tests for proving that the difference in the results is statistically significant. Researchers can find out from our study, how setting the number of subcomponents and choosing DE mutation schemes affect the performance of solving cLSGO.
Also, the manuscript is incomplete in the sense that figure 12 cannot be visualized - please address this issue and resubmit for re-evaluation
We may assume, this is a technical bug, because the figure is presented in DOCX version and it is presented in the similarity check report. Nevertheless, we carefully control all figures in updated version of our paper and we add all figures in supplementary materials.
Reviewer 2 Report
The paper proposes a new variant of ε-CC-SHADE metaheuristics for solving constraint large-scale global optimization (cLSGO) problems. The new variant is increasing size of groups of variables at the decomposition stage (iCC) and is called ε-iCC-SHADE. As such, novelty of this work is small. In the experimental part the ε-iCC-SHADE has been compared to the previously proposed ε-CC-SHADE with different population sizes and different numbers of subcomponents. The authors claim outperformance of the proposed ε-iCC-SHADE over ε-CC-SHADE.
1) The authors wrote: “In the first iterations, EA should use principles of global search. And at the end of the search process, EA should use methods of local search around the best found solutions.” These concepts are better known as exploration and exploitation of search space.
2) Pseudocode of the proposed ε-iCC-SHADE has been completely omitted.
3) I am not sure if comparison was fair. Namely, the authors wrote: “It is worth mentioning that, to evaluate some generation with m subcomponents and pop_size number of individuals, it is require to calculate m∙pop_size solutions.” So, when comparing ε-iCC-SHADE(1) and ε-iCC-SHADE(2) in the former case we have pop_size evaluations and in the latter case 2*pop_size evaluatuons. Furthermore, the number of generated solutions depend on pop_size as well. Authors experimented with the following population sizes: 25, 50, 75 and 100. In all of these different cases the number of fitness evaluations consumed must be MaxFEV. If the comparison is indeed fair the authors must explicitly discuss this issue in the paper.
4) In Figure 1 y-coordinate is strange. What numbers 1,2 … 7 stands for? Only later become clear that these numbers represent ranks. It would be much better if fitness values would be shown on y-coordinate.
5) Similarly, what are x-coordinates on Figure 2-5? From Figures 2-5 we can conclude that rank for particular metaheuristic is drastically changing over x-coordinate (might represents generations or particular independent run, but I am guessing). This is not a good sign. Can we rely on such changing information? Seems that last number on x-coordinate (18) is arbitrary selected and completely different results can be obtained if other setting is used.
6) Mann–Whitney U test has been used. Hence, family-wise error - type I error has been committed when performing multiple hypotheses tests. Hence, some post-hoc tests are needed or Friedman test.
7) Figure 12 is empty.
8) This paper is an extended version of the paper published in IOP Conference Series: Materials Science and Engineering, Volume 734, II International Scientific Conference "Advanced Technologies in Aerospace, Mechanical and Automation Engineering" - MIST: Aerospace - 2019 18–21 November 2019, Krasnoyarsk, Russia. I have not checked the conference version and if enough new material is indeed added. I do believe that this should be done by journal editor before submitting the paper to reviewers.
9) Typos:
IEEE Congress of Evolution Computation
->
IEEE Congress of Evolutionary Computation
titles as d ε-iCC-SHADE -> titles as ε-iCC-SHADE
References used in this review:
Črepinšek et al. 2013: Exploration and exploitation in evolutionary algorithms: A survey. ACM Comput. Surv. 45(3): 35, 2013.
Črepinšek et al. 2014: Replication and comparison of computational experiments in applied evolutionary computing: Common pitfalls and guidelines to avoid them. Applied Soft Computing, 19 (2014) 161–170.
Derrac et al. 2011: A practical tutorial on the use of nonparametric statistical tests as a methodology for comparing evolutionary and swarm intelligence algorithms, Swarm Evolut. Comput. 1 (1):3–18, 2011.
Author Response
The paper proposes a new variant of ε-CC-SHADE metaheuristics for solving constraint large-scale global optimization (cLSGO) problems.
The new variant is increasing size of groups of variables at the decomposition stage (iCC) and is called ε-iCC-SHADE.
As such, novelty of this work is small. In the experimental part the ε-iCC-SHADE has been compared to the previously proposed ε-CC-SHADE with different population sizes and different numbers of subcomponents.
The authors claim outperformance of the proposed ε-iCC-SHADE over ε-CC-SHADE.
First, we would like to thanks the reviewer for so detailed and very useful remarks.
1) The authors wrote: “In the first iterations, EA should use principles of global search. And at the end of the search process, EA should use methods of local search around the best found solutions.” These concepts are better known as exploration and exploitation of search space.
We have added the reference to the conception of exploration-and-exploitation in the test, when discuss idea of changing grouping sizes.
2) Pseudocode of the proposed ε-iCC-SHADE has been completely omitted.
We have added pseudocode of ε-iCC-SHADE in Table 1.
3) I am not sure if comparison was fair. Namely, the authors wrote: “It is worth mentioning that, to evaluate some generation with m subcomponents and pop_size number of individuals, it is require to calculate m∙pop_size solutions.” So, when comparing ε-iCC-SHADE(1) and ε-iCC-SHADE(2) in the former case we have pop_size evaluations and in the latter case 2*pop_size evaluatuons. Furthermore, the number of generated solutions depend on pop_size as well. Authors experimented with the following population sizes: 25, 50, 75 and 100. In all of these different cases the number of fitness evaluations consumed must be MaxFEV. If the comparison is indeed fair the authors must explicitly discuss this issue in the paper.
The comparison is fair. When we increase the number of subcomponent, we decrease the number of generations. Thus the MaxFEV remains the same in all experiments. This is described in the text in Section 4.2.
4) In Figure 1 y-coordinate is strange. What numbers 1,2 … 7 stands for? Only later become clear that these numbers represent ranks. It would be much better if fitness values would be shown on y-coordinate.
The comparison using ranks is more representative, because fitness values have different scales, and a EA in the worst run can get very bad and distinct from the best-found value. The rank-based comparisons are also used in many well-known benchmarks and competitions on evolutionary optimization from IEEE CEC and GECCO, including LSGO competitions.
5) Similarly, what are x-coordinates on Figure 2-5? From Figures 2-5 we can conclude that rank for particular metaheuristic is drastically changing over x-coordinate (might represents generations or particular independent run, but I am guessing). This is not a good sign. Can we rely on such changing information? Seems that last number on x-coordinate (18) is arbitrary selected and completely different results can be obtained if other setting is used.
These figures uses so-called ‘parallel diagram’ plots. We have modified these figures in order to combine separable and non separable problems, and we have added extended description in the text.
6) Mann–Whitney U test has been used. Hence, family-wise error - type I error has been committed when performing multiple hypotheses tests. Hence, some post-hoc tests are needed or Friedman test.
We have performed extended analysis using post-hoc test of Holm. The results are shown in Tables 7 and 9.
7) Figure 12 is empty.
We may assume, this is a technical bug, because the figure is presented in DOCX version and it is presented in the similarity check report. Nevertheless, we carefully control all figures in updated version of our paper and we add all figures in supplementary materials.
8) This paper is an extended version of the paper published in IOP Conference Series: Materials Science and Engineering, Volume 734, II International Scientific Conference "Advanced Technologies in Aerospace, Mechanical and Automation Engineering" - MIST: Aerospace - 2019 18–21 November 2019,
Krasnoyarsk, Russia. I have not checked the conference version and if enough new material is indeed added. I do believe that this should be done by journal editor before submitting the paper to reviewers.
We have carried out new experiments, provided extended analysis of the results. We have stated and checked a new hypothesis about mutation strategies in our algorithms.
The paper contains many new data, plots, tables and conclusions.
9) Typos:
IEEE Congress of Evolution Computation
->
IEEE Congress of Evolutionary Computation
titles as d ε-iCC-SHADE -> titles as ε-iCC-SHADE
We have fixed these typos.
References used in this review:
Črepinšek et al. 2013: Exploration and exploitation in evolutionary algorithms: A survey. ACM Comput. Surv. 45(3): 35, 2013.
Črepinšek et al. 2014: Replication and comparison of computational experiments in applied evolutionary computing: Common pitfalls and guidelines to avoid them. Applied Soft Computing, 19 (2014) 161–170.
Derrac et al. 2011: A practical tutorial on the use of nonparametric statistical tests as a methodology for comparing evolutionary and swarm intelligence algorithms, Swarm Evolut. Comput. 1 (1):3–18, 2011.
Reviewer 3 Report
This paper presents an improved version of a cooperative coevolution framework for solving constrained large-scale global optimization problems. The framework integrates cooperative coevolution to deal with high dimensionality, a variant of differential evolution to perform the actual optimization and a strategy borrowed from epsilon-differential evolution to deal with the problems constraints. The main contribution of this work is the investigation of the effect of increasing the the number of variables in subcomponents during the algorithm run instead of using fixed sized variable subsets.
While I found the text reasonably clear, there are many typos, concordance problems, words used in strange contexts and several other issues that must be addressed by a thorough revision of the paper done by a native speaker or someone with similar domain of the english language.
Overall, I found the approach described interesting, and the experimental results seem to confirm the superiority of this version of the algorithm when compared with the previous version, where the size of the subsets of variables was fixed. There are however two main issues with this paper that I think should be addressed for it to be published.
- Firstly, I think the algorithm should also be compared to one or more other state of the art algorithms in this area, including non-decomposition methods, to really evaluate if this approach is a relevant contribution to this field or simply a minor increment to a pre-existing algorithm. I believe a marginal improvement over a similar algorithm hardly justifies a journal publication, even when considering the additional study of different mutation operators.
- Secondly, I would like to see a discussion on how the algorithm performs on the separable vs. non separable functions in the test set, since I believe previous critiques on the cooperative coevolution approach usually where related to its difficulty in dealing with non-separable problems where the intense interaction between variables make it difficult to optimize subsets of variables independently. There is inclusively significant work on strategies to move variables between subsets so that related variables are optimized together.
Author Response
This paper presents an improved version of a cooperative coevolution framework for solving constrained large-scale global optimization problems. The framework integrates cooperative coevolution to deal with high dimensionality, a variant of differential evolution to perform the actual optimization and a strategy borrowed from epsilon-differential evolution to deal with the problems constraints. The main contribution of this work is the investigation of the effect of increasing the the number of variables in subcomponents during the algorithm run instead of using fixed sized variable subsets.
First, we would like to thanks the reviewer for so detailed and very useful remarks.
While I found the text reasonably clear, there are many typos, concordance problems, words used in strange contexts and several other issues that must be addressed by a thorough revision of the paper done by a native speaker or someone with similar domain of the english language.
Overall, I found the approach described interesting, and the experimental results seem to confirm the superiority of this version of the algorithm when compared with the previous version, where the size of the subsets of variables was fixed.
There are however two main issues with this paper that I think should be addressed for it to be published.
- Firstly, I think the algorithm should also be compared to one or more other state of the art algorithms in this area, including non-decomposition methods, to really evaluate if this approach is a relevant contribution to this field or simply a minor increment to a pre-existing algorithm. I believe a marginal improvement over a similar algorithm hardly justifies a journal publication, even when considering the additional study of different mutation operators.
The investigated class of optimization problems, namely cLSGO, is rather new and not well studied. We believe that we are one of the first who have discussed and proposed new algorithms for intersection of LSGO and constrained global optimization. At least, we have not found any referenced to cLSGO in resent publications in the field (IEEE CEC, GECCO, PPSN and other conference).
Our experience in solving not constrained LSGO shows that many state-of-the-art techniques for not LSGO problems perform very bad and usually can’t obtain any acceptable solution. Thus, we investigate only those approaches that are specially designed for cLSGO.
- Secondly, I would like to see a discussion on how the algorithm performs on the separable vs. non separable functions in the test set, since I believe previous critiques on the cooperative coevolution approach usually where related to its difficulty in dealing with non-separable problems where the intense interaction between variables make it difficult to optimize subsets of variables independently. There is inclusively significant work on strategies to move variables between subsets so that related variables are optimized together.
We have performed extended analysis for separable and non-separable problems for e-CC-SHADE (m) variants and e-iCC-SHADE algorithm. The results are presented in 4.3 and 4.4 sections.
Round 2
Reviewer 2 Report
Still issues which should be fixed remain.
1) The authors wrote in the responding letter: “We have added the reference to the conception of exploration-and-exploitation in the test, when discuss idea of changing grouping sizes”. But, references to exploration and exploitation were not included. Also, references should be used in the beginning of Section 3 where exploration and exploitation have been discussed.
2) Pseudocode should be without “goto” statements. Goto statements support “spaghetti” code and have been considered harmful in software engineering since 1970. Provided pseudocode is not adequate.
3) The authors wrote: “The comparison is fair. When we increase the number of subcomponent, we decrease the number of generations. Thus the MaxFEV remains the same in all experiments. This is described in the text in Section 4.2.” So, they are still using termination based on maximum number of iterations/generations. But, in the paper authors claim that the termination is based on maximum number of fitness evaluations. Hence, actual implementation does not match what is described in the paper. Would be good if authors can provide a link to actual implementation so that can be verified a reviewers/readers. I have seen to many descriptions of metaheuristics which do not match actual implementation.
4) Figure 21 is still empty. Please check generated pdf.
5) Typo
Figure3 demonstrate -> Figure 3 demonstrate
6) Some results of M-W U tests are in tables (e.g., Table 5), others in figures (e.g., Figure 18).
Author Response
We would like to once again thank the reviewer for the informative comments.
1) The authors wrote in the responding letter: “We have added the reference to the conception of exploration-and-exploitation in the test, when discuss idea of changing grouping sizes”. But, references to exploration and exploitation were not included. Also, references should be used in the beginning of Section 3 where exploration and exploitation have been discussed.
We have added the references proposed in the first review ([25] Črepinšek, M.; Liu, S.-H.; Mernik, M. Exploration and exploitation in evolutionary algorithms: A survey. ACM Computing Surveys 2013, 45(3), 1–33. doi:10.1145/2480741.2480752.)
2) Pseudocode should be without “goto” statements. Goto statements support “spaghetti” code and have been considered harmful in software engineering since 1970. Provided pseudocode is not adequate.
We have changed the pseudocode in order to present the algorithm without “goto”. Also, we would like to add that do not use “goto” in our software implementations. We assumed that this “old style” in more clear for those who are not a software designer.
3) The authors wrote: “The comparison is fair. When we increase the number of subcomponent, we decrease the number of generations. Thus the MaxFEV remains the same in all experiments. This is described in the text in Section 4.2.” So, they are still using termination based on maximum number of iterations/generations. But, in the paper authors claim that the termination is based on maximum number of fitness evaluations. Hence, actual implementation does not match what is described in the paper. Would be good if authors can provide a link to actual implementation so that can be verified a reviewers/readers. I have seen to many descriptions of metaheuristics which do not match actual implementation.
We will explain the termination criteria in detail. First, we do not use “termination based on maximum number of iterations/generations”, but we point out in the text: “The termination criterion is the FEV budget exhaustion”. This termination criterion is the same as in the IEEE CEC competitions and it models the real-world applications when we have strong limitations in the number of evaluations of an objective function.
All our EAs operate with populations and change them from generation to generation. This means that for creating new candidate-solutions (new generation) we use a set of solutions (the previous generation). After that, new solutions should be evaluated. At the evaluation stage, we control the budget of FEVs. If maxFEV is reached, we stop our algorithm, and we use the best-found solution from all previous evaluations as the solution to the problem. We can stop the algorithm, even we have evaluated only a part of the current population.
Thus we use iterations/generations as an internal parameter of the control of evolution. But the termination criterion is maxFEV.
As about our implementation. We do not design a user-friendly software, because we need only a framework for numerical experiments and for investigating the performance of approaches. Moreover, the experiments are computationally expensive and take many days of run-time. Our software is designed for applying parallel computations adapted for our cluster-PC with multi-core processors. Nevertheless, we supply you with the github link (https://github.com/pertyzxxx/Algorithms_Journal), and the general logic of our algorithms and experiments can be found there.
4) Figure 21 is still empty. Please check generated pdf.
We have carefully checked the content of our DOCX and PDF files when submitted them, and all pictures were ok. We assume that the submission system generates PDF online from DOCX file. We have changed picture formatting inside DOCX and we hope the issue will not appear again. Nevertheless, we have attached all source pictures in the zip archive.
5) Typo
Figure3 demonstrate -> Figure 3 demonstrate
The typo is corrected.
6) Some results of M-W U tests are in tables (e.g., Table 5), others in figures (e.g., Figure 18).
Now we use one style (namely, tables).

Reviewer 3 Report
I believe the issues pointed out in the previous review are sufficiently addressed for the paper to be published after a minor revision:
- The english could still be a bit improved, please perform an additional review to improve the language issues pointed out previously.
- There is a space where image 21 should be, I'me not sure if it is a problem on my side.
- The conclusion section could be a bit more fleshed out to better summarize the contributions made by the paper.
Author Response
We would like to once again thank the reviewer for the informative comments.
I believe the issues pointed out in the previous review are sufficiently addressed for the paper to be published after a minor revision:
1) The english could still be a bit improved, please perform an additional review to improve the language issues pointed out previously.
We carefully checked the text of the article and have found and corrected many errors and typos.
2) There is a space where image 21 should be, I'me not sure if it is a problem on my side.
We have carefully checked the content of our DOCX and PDF files when submitted them, and all pictures were ok. We assume that the submission system generates PDF online from DOCX file. We have changed picture formatting inside DOCX and we hope the issue will not appear again. Nevertheless, we have attached all source pictures in the zip archive.
3) The conclusion section could be a bit more fleshed out to better summarize the contributions made by the paper.
We have revised the conclusion section and have added more details on the summary of the results.
